# Unifying Cross-Lingual Transfer across Scenarios of Resource Scarcity

**Alan Ansell**[*1]   **Marinela Parović**[*1]   **Ivan Vulić**[1]
**Anna Korhonen**[1]   **Edoardo Maria Ponti**[2,1]
[1]Language Technology Lab, University of Cambridge
[2]University of Edinburgh
{aja63,mp939}@cam.ac.uk

## Abstract

The scarcity of data in many of the world's languages necessitates the transfer of knowledge from other, resource-rich languages. However, the level of scarcity varies significantly across multiple dimensions, including: i) the amount of task-specific data available in the source and target languages; ii) the amount of monolingual and parallel data available for both languages; and iii) the extent to which they are supported by pretrained multilingual and translation models. Prior work has largely treated these dimensions and the various techniques for dealing with them separately; in this paper, we offer a more integrated view by exploring how to deploy the arsenal of cross-lingual transfer tools across a range of scenarios, especially the most challenging, low-resource ones. To this end, we run experiments on the Americas-NLI and NusaX benchmarks over 20 languages, simulating a range of few-shot settings. The best configuration in our experiments employed parameter-efficient language and task adaptation of massively multilingual Transformers, trained simultaneously on source language data and both machine-translated and natural data for multiple target languages. In addition, we show that pre-trained translation models can be easily adapted to unseen languages, thus extending the range of our hybrid technique and translation-based transfer more broadly. Beyond new insights into the mechanisms of cross-lingual transfer, we hope our work will provide practitioners with a toolbox to integrate multiple techniques for different real-world scenarios. Our code is available at https://github.com/parovicm/unified-xlt.

## 1 Introduction

While training data is available for a wide range of NLP tasks in a handful of high-resource languages, the vast majority of the world's languages with their billions of speakers do not fit into this category (Joshi et al., 2020; Blasi et al., 2022). The lack of labelled data makes it difficult or impossible to directly train effective NLP systems for these languages. For this reason, researchers have looked for ways to harness data from one or more high-resource "source" languages to compensate for a shortage of data in low-resource "target" languages in a process known as "cross-lingual transfer" (XLT). Different techniques have been developed to deal with the various dimensions of resource scarcity, which encompass not just data availability, but also the degree of support by pretrained models. These research threads have generally been investigated somewhat independently. In this paper, we attempt to unify several of the most prominent threads of XLT research into a single framework. Specifically, we synthesise findings from zero-shot (ZS) XLT with massively multilingual transformers (MMTs), few-shot (FS) XLT, XLT for low-resource languages and XLT through machine translation (MT) to formulate a practical, general-purpose approach to cross-lingual transfer, with a focus on low-resource scenarios.

Massively multilingual Transformers (MMTs), Transformer-based architectures (Vaswani et al., 2017) pretrained with an unsupervised objective such as masked language modelling (MLM) on text from a large number of languages, are perhaps the most fundamental tool for contemporary XLT. Prominent examples of MMTs include mBERT (Devlin et al., 2019), XLM-R (Conneau et al., 2020a) and mDeBERTa (He et al., 2023). In addition to providing broad language coverage, MMTs have been shown to learn representations which have a degree of cross-lingual alignment, even though they do not receive any explicit cross-lingual signal during training (Conneau et al., 2020b; Muller et al., 2021). This allows an MMT fine-tuned for a specific task in a given "source" language to perform the same task in another "target" language with a level of performance generally

---
*Equal contribution.

much better than random chance (Pires et al., 2019; Wu and Dredze, 2019), despite never having seen a single example of the task in the target language; this is known as "zero-shot" cross-lingual transfer.

While MMTs typically cover around 100 languages, this is still only a small fraction of the world's estimated 7,000 languages. Pfeiffer et al. (2020) and Ansell et al. (2022) have shown that an effective strategy for ZS-XLT for target languages not covered by the MMT is to learn a parameter-efficient fine-tuning (PEFT) to specialise the MMT to that "unseen" language. The resulting language module can be composed with a task module, typically yielding much better performance than the MMT could achieve without language adaptation.

In contrast to ZS-XLT, often more realistic is the few-shot case (FS-XLT), where a small number of gold-standard target language examples are available during training. Though it may be expensive to annotate target language data, especially for low-resource languages where native speakers are hard to access, prior work has shown that using even a small amount during training can yield significant gains in performance (Zhao et al., 2021). While early approaches to FS-XLT involved fine-tuning first on the source language data, then separately on the few target language shots (Lauscher et al., 2020), recent work has shown that it is more effective to jointly train on both at once (Xu and Murray, 2022; Schmidt et al., 2022).

Another tool often employed for cross-lingual transfer is machine translation (MT). MT approaches can generally be categorised as *translate-train* or *translate-test* (Hu et al., 2020). To confine the scope of our work, we consider only the translate-train approach, which has so far been predominant, although Artetxe et al. (2023) have made a strong case for considering translate-test further in future work. We consider two translate-train variants: TTRAIN-SINGLE, where a model is trained for each target language using only its own translated data; and TTRAIN-ALL, where one model covering all target languages is trained on their translated data and the source language data simultaneously.

In this work, we consider how best to employ the above techniques in response to cross-lingual transfer scenarios with varying levels of data scarcity. We thus explore several promising directions of integrating the zero-shot, few-shot and translate-train techniques across a range of resource levels in order to delve deeper into: (i) to what extent these techniques and the different data sources that they exploit are complementary, (ii) what is the most effective way of combining different data sources in order to maximise the performance, and (iii) how much each of the available sources of data contributes to the overall performance. We aim to equip practitioners with a recipe for how to use the available data resources in the most effective way.

We experiment on the AmericasNLI natural language inference (NLI) dataset for American languages (Ebrahimi et al., 2022), and the NusaX sentiment analysis (SA) dataset for Indonesian languages (Winata et al., 2023). We find that combining language adaptation, few-shot learning and translation can be highly effective, yielding average performance gains of 14-24 points over the zero-shot baseline without language adaptation.

## 2 Methodology

### 2.1 Background

Because MMTs divide their capacity among many languages, they may often perform sub-optimally with respect to a single source or target language. Furthermore, we are sometimes interested in a target language not covered by the MMT. A naive solution to these problems is to prepare the MMT with continued pretraining on the target language before proceeding to task fine-tuning. While this can improve performance, Pfeiffer et al. (2020) and Ansell et al. (2022) show that a more effective approach is to apply a form of composable, parameter-efficient fine-tuning during continued pretraining: Pfeiffer et al. (2020) employ *adapters* (Rebuffi et al., 2017; Houlsby et al., 2019), while Ansell et al. (2022) propose sparse fine-tunings (SFTs) learned through an algorithm they call "Lottery Ticket Sparse Fine-Tuning" (LT-SFT). The resulting language-specific module ("language module") can be composed with a similar module trained for the task of interest ("task module") to perform zero-shot transfer. This approach is not only more efficient than sequential full fine-tuning in terms of model size and training time; keeping the MMT weights frozen while training the language modules helps prevent the model from forgetting important knowledge learned during pretraining.

While SFT composition generally exhibits somewhat better zero-shot cross-lingual transfer performance across a range of tasks than adapter composition (Ansell et al., 2022; Alabi et al., 2022), and avoids the overhead incurred during inference by

adapters, adapters are more efficient to train and leave the base MMT fully unmodified. In this work, we consider both methods. However, we note that other modular, parameter-efficient methods exist and could be used with our tools in future work (Pfeiffer et al., 2023).

**Multi-Source Training.** Multi-source training is an extension to parameter-efficient language adaptation, where a task adapter is trained using data from several source languages simultaneously, often yielding large gains in cross-lingual transfer performance as a result of the task adapter learning more language-agnostic representations (Ansell et al., 2021, 2022). Multi-source training requires that each training batch consists of examples from the same language so that the module for the relevant language can be applied during each step.

## 2.2 Recipe

We propose a recipe for cross-lingual transfer which is flexible and effective across scenarios of resource scarcity. It can be summarised as:

1. Select a base MMT and train language modules for the source language and all target languages for which monolingual data is available.
2. Using a multilingual NMT model, translate the task data into every target language it supports; if there are target languages the MT model does not support but for which parallel data is available, adapt it using this parallel data.
3. Learn a task module through joint multi-source training on all available data (i.e. source data, translated data and any gold-standard target language ("few-shot") data available).

We propose two methodological novelties to enhance this recipe.

**Few-shot upsampling.** When gold-standard target language data is available during training, it is generally present in a much smaller quantity than the source language data (in our case, the difference is in orders of magnitude, but it can vary). Furthermore, it is typically higher in quality than machine-translated target language data. For this reason, we suggest upsampling this few-shot data relative to the source and machine-translated data during multi-source training. We show in our experiments that this can improve downstream performance.

**NMT model adaptation.** While recent multilingual NMT models such as NLLB (NLLB Team et al., 2022) provide impressive language coverage, there are still many languages they do not

support. We therefore adapt NLLB to unseen target languages by initialising a new language token and embedding for the target language and then performing continued pretraining with parallel data for the relevant language pair.[1]

# 3 Experimental Setup

## 3.1 Evaluation Tasks and Languages

We evaluate our models on two classification tasks: natural language inference (NLI) and sentiment analysis (SA). For NLI, we use the AmericasNLI dataset (Ebrahimi et al., 2022), which covers 10 low-resource languages from the Americas. For SA, we opt for the NusaX dataset (Winata et al., 2023), spanning 10 low-resource Indonesian languages. In the NLI task, the source language is English, while for SA it is Indonesian. We provide the list of all datasets and languages used in Table 1. These tasks are particularly amenable to the translate-train approach since the labels are preserved even after data has been translated into another language. The complete overview of the languages and their codes is given in Appendix A.

## 3.2 Models and Training Details

**MMT.** In this work, we use the base version of XLM-R (Conneau et al., 2020a), an MMT with 270M parameters pretrained on 100 languages.[2]

**NMT model.** As our primary MT model for obtaining translated data, we choose the NLLB model with 3.3B parameters (NLLB Team et al., 2022), trained to translate between any pair of 200+ languages, including many low-resource languages. We also experiment with two additional NLLB variants: distilled models with 600M and 1.3B parameters, enabling us to understand the effect of model size on the "quality" of the obtained data. Despite the broad language coverage, half of our target languages are unsupported by the NLLB models (7

---

[1]In fact, it is not necessary for the source language in the parallel corpus to match the intended source language for cross-lingual transfer, since multilingual NMT models can, in theory, support unseen transfer directions provided the source and target languages have been seen as part of other pairs during training; this is the case in NMT adaptation for AmericasNLI, where the parallel data is Spanish-to-X but the transfer direction is English-to-X.

[2]While more powerful MMTs are available, such as XLM-R-large or mDeBERTa (He et al., 2023), our primary purpose is not the maximisation of raw performance, nor a comparison of different MMTs, so we opt for a smaller model to stretch our computational budget over a broad range of scenarios and languages.

| Task | Target Dataset(s) | Source Dataset(s) | MMT | Target Languages |
|---|---|---|---|---|
| Natural Language Inference (NLI) | AmericasNLI (Ebrahimi et al., 2022) (sh: 743 / tst: 750) | MultiNLI (tr: 393K / dev: 10K) (Williams et al., 2018) | XLM-R Base | Aymara*, Asháninka*†, Bribri*†, Guarani*, Náhuatl*†, Otomí*†, Quechua*, Rarámuri*†, Shipibo-Konibo*†, Wixarika*† |
| Sentiment Analysis (SA) | NusaX (Winata et al., 2023) (sh: 600, tst: 400) | SMSA (tr: 11K, dev: 1.3K) (Purwarianti and Crisdayanti, 2019; Wilie et al., 2020) | XLM-R Base | Acehnese*, Balinese*, Banjarese*, Buginese*, Javanese, Madurese*†, Minangkabau*, Ngaju*†‡, Sundanese, Toba Batak*† |

Table 1: Details of tasks, datasets, MMTs and languages involved in our experiments. sh = # of few-shot examples available per target language; tst = # of test set examples; tr = # of train set examples; dev = # of development set examples; * denotes languages unseen during MMT pretraining; † denotes languages not supported by the NLLB MT model; ‡ denotes languages for which no satisfactory monolingual corpus was available and hence no language module was trained. Further details of all the language and data sources used are provided in Appendix A. Note that since the NusaX dataset is created through human translation of a subset of the SMSA dataset, we carefully remove every example from SMSA which appears in its original or modified form in the NusaX test set to avoid a data leak.

languages from the AmericasNLI and 3 languages from NusaX dataset).

We adapt the 3.3B parameter NLLB model to unseen languages through continued pretraining on the parallel corpora listed in Appendix A. We perform full fine-tuning for 5 epochs with a batch size of 8 and an initial learning rate of $2 \cdot 10^{-5}$ which is linearly decreased to zero during training.

**Language Modules.** In general, we use the same algorithms and hyperparameters as the original papers (Pfeiffer et al., 2020; Ansell et al., 2022) when training language modules. However, we use the variant of MAD-X proposed by Pfeiffer et al. (2021), where the last adapter layers are dropped for an increase in cross-lingual transfer performance. We provide a list of resources for the monolingual corpora in Appendix A. Language modules are trained for a minimum of 100 epochs and 100,000 steps with a batch size of 8, a learning rate of $5 \cdot 10^{-5}$ and a maximum sequence length of 256. We evaluate the language modules every 1,000 steps with low-resource languages, and every 5,000 steps with high-resource languages. Finally, we choose the module that has obtained the lowest perplexity on the validation set, which is created by taking 5% of the unlabelled data for low-resource languages or 1% for high-resource languages.

**Task Modules.** We again follow Pfeiffer et al. (2020) and Ansell et al. (2022) except where stated otherwise. We train task adapters with a reduction factor of 16 (i.e. the ratio between the dimension of the MMT hidden state and the dimension of the adapter hidden state is 16) and task SFTs with 8% density. When jointly training on data from

more than one language, the training examples are batched such that each batch consists of examples from a single language, and the batches are ordered randomly. For the configurations which employ language adaptation, the language module for the relevant language is activated at the beginning of the training step and deactivated at the end of the step, following Ansell et al. (2021).

AmericasNLI task modules are trained for 5 epochs with a batch size of 32 and an initial learning rate of $2 \cdot 10^{-5}$. Evaluation is carried out every 625 steps and the checkpoint with the best evaluation accuracy is selected at the end of training. NusaX task modules are trained for 10 epochs (or 3 during the full fine-tuning phase of LT-SFT), with a batch size of 16 and an initial learning rate of $2 \cdot 10^{-5}$. They are evaluated after every 250 steps and the final module is the one with the best evaluation F1 score. For both tasks, the learning rate is linearly decreased to zero over the course of training.

### 3.3 Configurations and Ablations

**ZS-XLT.** We include zero-shot transfer results with language adaptation, equivalent to MAD-X (Pfeiffer et al., 2020) in the case of adapters. We also have a variant where language adaptation is not employed, thus only the task module is used for training and inference. These variants are denoted by ZS and ZS − LA, respectively.

**FS-XLT.** In our default FS-XLT setup, "FS-SINGLE" we add $K = 100$ target shots to the source language task data, training a separate task module for each target language. We also consider "FS-ALL", where a single task module is trained

on the source language data plus $K = 100$ shots from each target language. We investigate the effect of different numbers of shots by also carrying out FS-SINGLE experiments with $K \in \{20, 500\}$.[3] We employ language adaptation in all these setups, but as an ablation, we also test $K = 100$ without language adaptation (denoted as FS − LA).

In all FS experiments, the model is jointly trained on source and target data, as per Xu and Murray (2022). We upsample the data in the target language(s) by a factor of 10 to increase its presence during training, since $K$ is still rather small compared to the number of examples available in the source language. During training, we only evaluate on the source language data following Xu and Murray (2022), who point out that the presence of large evaluation sets in truly low-resource languages is unrealistic[4], and show that while evaluating on the target language is still beneficial for the joint training procedure, the gap becomes much smaller. They stress that such data would be better used for training, in line with Kann et al. (2019).

**Translate-Train.** In our main translate-train variant, named TTRAIN-ALL, we create a single task module covering all the target languages, which is trained and evaluated on the translated data of all target languages together with the source language data. We also consider TTRAIN-SINGLE, where a separate task module is trained on the data of each target language alone.

**FS-XLT meets Translate-Train.** In a final set of experiments, we investigate to what extent the benefits gained from the few-shot and translate-train methods add up when they are combined. To test this, we introduce the FS + TTRAIN-ALL configuration, where we train a single task module on the union of the source language data and translated and few-shot data (with $K = 100$) for every target language. This module is evaluated on the source language data and the translated data in all target languages.

## 4 Results and Discussion

**Main Results.** The results of our primary configurations on NLI and SA are presented in Table 2,

with ablations shown in Table 3.[5] We find that the various cross-lingual transfer techniques we consider can be combined very effectively to improve performance. For instance, the average SA performance can be improved from the most basic ZS − LA setting by 14-17 points (depending on the PEFT method) through the use of language adaptation, translate-train and few-shot techniques with $K = 100$ shots (FS + TTRAIN-ALL). In the case of the NLI task, the gains under the same conditions are 19-24 points. Each of these components individually adds several points of performance, and although the gain from using FS and TTRAIN together is much smaller than the sum of their individual gains, it is still 1-2 points better than using either technique on its own. Although we did not consider FS + TTRAIN-ALL with $K = 500$ shots, the strength of FS-SINGLE with $K = 500$ as shown in Figure 1 suggests that this gap would be larger with larger $K$. The finding that high-quality machine translation of the entire source dataset cannot eliminate the utility of human-crafted examples contains a potentially useful lesson – we would encourage designers of datasets for cross-lingual transfer to provide at least two splits for target languages, even if the training/validation split contains only 100 examples. The relative value of few-shot and machine-translated data appears to be task-dependent. Whereas for AmericasNLI we see TTRAIN outperforming FS by 4-6 points, neither approach has a clear advantage on the NusaX task.

**MT Model Size.** In Table 4, we see the effect of translation model quality on TTRAIN-ALL performance, with gains of 0.5-2 points from upscaling the NLLB model from 600 million to 3.3 billion parameters. This upscaling comes at a relatively small cost in the translate-train setup, since the training data only needs to be translated once for each target language. Translate-test setups, on the other hand, incur the cost of translating each example encountered at inference time, which is potentially much more costly for large-scale deployments.

**MT adaptation.** The adaptation of the NMT model to new languages appears to be highly effective, with these languages generally enjoying large gains from the use of TTRAIN despite the small size of the parallel corpora available: the AmericasNLI

---

[3]For the AmericasNLI target languages NAH and OTO we actually use 223 and 377 shots respectively under the $K = 500$ setting since this is the maximum available.

[4]The requirement for evaluation data in the target language originated from two-step methods, where such data is needed to prevent the model from overfitting to the small number of examples in the target language used during the second stage.

---

[5]For Nusa-X, our main results exclude NIJ for which no monolingual data is available and thus no language module was trained; results without language adaptation are available in Appendix B.

| | Method | AYM | BZD * | CNI * | GN | HCH * | NAH * | OTO * | QUY | SHP * | TAR * | **avg** |
|---|---|---|---|---|---|---|---|---|---|---|---|---|
| **ADAPTER** | ZS | 53.0 | 42.8 | 44.8 | 59.6 | 40.3 | 50.8 | 41.2 | 55.2 | 50.0 | 40.0 | 47.77 |
| | FS-SINGLE | 55.2 | 47.1 | 47.7 | 58.3 | 40.7 | 55.8 | 45.9 | 59.7 | 52.9 | 47.5 | 51.08 |
| | FS-ALL | 57.1 | 47.9 | 49.3 | 59.1 | 44.1 | 54.5 | **48.4** | 59.6 | 52.3 | 47.5 | 51.98 |
| | TTRAIN-SINGLE | 58.7 | 56.5 | 53.9 | 64.4 | 48.7 | 56.4 | 40.1 | 61.5 | 57.5 | 51.5 | 54.92 |
| | TTRAIN-ALL | 63.3 | **60.5** | 56.1 | 66.1 | **52.1** | **60.4** | 42.4 | **64.9** | 63.3 | 55.3 | 58.44 |
| | FS + TTRAIN-ALL | **63.6** | 59.1 | **57.7** | 67.3 | **52.1** | 59.6 | 48.3 | 64.5 | **64.4** | **56.7** | **59.33** |
| **SFT** | ZS | 58.4 | 44.7 | 47.6 | 62.2 | 44.4 | 50.8 | 46.4 | 60.4 | 49.5 | 43.1 | 50.75 |
| | FS-SINGLE | 59.2 | 58.7 | 53.2 | 63.9 | 45.9 | 54.6 | 49.1 | 61.2 | 53.1 | 51.2 | 55.01 |
| | FS-ALL | 60.8 | 58.1 | 52.3 | 63.3 | 47.3 | 56.5 | 53.3 | 61.3 | 54.9 | 51.3 | 55.91 |
| | TTRAIN-SINGLE | 64.5 | 58.7 | 54.9 | 69.2 | 53.2 | 61.5 | 43.4 | 65.1 | 59.5 | 56.1 | 58.61 |
| | TTRAIN-ALL | 65.5 | 61.9 | 56.7 | **70.5** | 55.5 | 62.2 | 42.6 | 68.5 | 66.0 | 58.7 | 60.81 |
| | FS + TTRAIN-ALL | 65.1 | 62.1 | 58.5 | 70.0 | 54.8 | 61.8 | 51.2 | 68.8 | 67.6 | 60.3 | 62.02 |

(a) AmericasNLI: accuracy

| | Method | ACE | BAN | BBC * | BJN | BUG | JAV | MAD * | MIN | SUN | **avg** |
|---|---|---|---|---|---|---|---|---|---|---|---|
| **ADAPTER** | ZS | 74.9 | 78.0 | 72.3 | 77.6 | 57.6 | 82.9 | 68.5 | 79.9 | 80.5 | 74.69 |
| | FS-SINGLE | 79.1 | 80.5 | 77.2 | 86.0 | 72.0 | 85.0 | 77.5 | 84.8 | 83.7 | 80.64 |
| | FS-ALL | 79.5 | 80.0 | 75.3 | **86.2** | 70.4 | **86.2** | 76.8 | 84.4 | 82.4 | 80.13 |
| | TTRAIN-SINGLE | 74.0 | 77.7 | 73.8 | 82.0 | 66.6 | 83.6 | 70.8 | 78.0 | 79.7 | 76.24 |
| | TTRAIN-ALL | 79.6 | 82.3 | 81.0 | 85.0 | 68.1 | **86.2** | 78.8 | 83.3 | **85.7** | 81.11 |
| | FS + TTRAIN-ALL | 82.3 | 82.9 | 82.8 | 84.9 | **72.2** | 85.3 | 80.5 | 85.7 | 85.0 | **82.40** |
| **SFT** | ZS | 80.0 | 81.3 | 65.8 | 82.0 | 63.8 | 84.3 | 73.5 | 86.6 | 84.4 | 77.97 |
| | FS-SINGLE | 82.2 | 84.1 | 80.6 | 88.3 | 77.7 | 88.0 | 78.6 | 89.2 | 85.1 | 83.76 |
| | FS-ALL | 83.7 | 87.2 | 79.6 | 87.9 | 75.9 | 87.3 | 77.2 | 86.7 | 84.0 | 83.30 |
| | TTRAIN-SINGLE | 82.4 | 82.0 | 83.5 | 85.4 | 68.1 | 85.6 | 80.9 | 87.0 | 83.0 | 81.93 |
| | TTRAIN-ALL | 83.4 | 82.4 | 82.7 | 84.6 | 76.8 | 86.4 | 81.6 | 87.4 | 85.2 | 83.39 |
| | FS + TTRAIN-ALL | 85.0 | 85.8 | 82.7 | 87.4 | 78.7 | 87.4 | 81.0 | 88.8 | 86.5 | 84.81 |

(b) NusaX: F1

Table 2: Results of ZS, FS and TTRAIN methods on AmericasNLI and NusaX with adapters and SFTs. For the FS methods, K = 100, and for the TTRAIN methods, the MT model is NLLB with 3.3B parameters. The last column is the average score over all languages. **Bold**: the best approach within Adapter/SFT. Underline: overall best score. *: NLLB MT model adaptation required to support this language.

| | ADAPTER | | SFT | |
|---|---|---|---|---|
| Method | NLI | SA | NLI | SA |
| ZS | **47.77** | **74.69** | 50.75 | 77.97 |
| ZS – LA | 40.57 | 65.38 | 38.21 | 70.43 |
| FS | **51.08** | **80.64** | 55.01 | 83.76 |
| FS – LA | 44.82 | 74.03 | 49.51 | 78.32 |
| FS – UPSAMPLE | 48.27 | 76.78 | 52.60 | 83.18 |
| TTRAIN-ALL | **58.44** | **81.11** | 60.81 | 83.39 |
| TTRAIN-ALL – LA | 55.50 | 77.83 | 57.88 | 79.39 |

Table 3: Ablation experiments: – LA denotes the corresponding method without language adaptation (i.e. only task module is present); – UPSAMPLE indicates no upsampling of the gold-standard target shots is performed. FS refers to the -SINGLE variant. The scores are averages over all target languages.

languages have parallel corpora containing 5,000-17,000 sentences, while the NusaX parallel corpora have fewer than 1,000 sentences. It is interesting to note that for AmericasNLI, the language pairs used in MT model adaptation are different from those used during cross-lingual transfer: the source language in the parallel corpora is Spanish, so the English-to-X direction required during translation of the MultiNLI dataset is completely unseen. The success of the TTRAIN configurations on this task is thus a testament to the strength and flexibility of multilingual NMT.

**Number of Shots.** We observe a rather large impact on performance from increasing the amount of few-shot data. While even 20 shots are enough to bring about a 3-5 point average gain on the NusaX task, we do not see a plateau in performance on either task even with the increase from 100 to 500 shots. Upsampling the few shots seems beneficial, yielding a 0.5-4 point gain in performance when $K = 100$. A finer-grained and wider exploration of this finding is warranted in future work.

**Language Resourcefulness.** As suggested by prior work (Pfeiffer et al., 2020; Ansell et al., 2021) and by Table 3, language adaptation has a very large impact on all configurations. In the case of the

| | Model | NLI | | | | SA | | | | | | | |
|---|---|---|---|---|---|---|---|---|---|---|---|---|---|
| | | AYM | GN | QUY | **avg** | ACE | BAN | BJN | BUG | JAV | MIN | SUN | **avg** |
| ADAPTER | NLLB 600M | **62.9** | 65.1 | 63.2 | 63.73 | 76.6 | 79.0 | 83.7 | 66.0 | 80.9 | 81.5 | 81.1 | 78.40 |
| | NLLB 1.3B | 62.3 | **67.9** | 63.2 | 64.47 | 73.4 | 80.2 | 85.3 | 64.8 | 83.4 | 80.6 | 83.0 | 78.67 |
| | NLLB 3.3B | 62.4 | 67.6 | 65.2 | **65.07** | **79.6** | 82.4 | 84.6 | 66.4 | 84.8 | 81.9 | 84.6 | **80.61** |
| SFT | NLLB 600M | 67.3 | 69.7 | 68.7 | 68.57 | 83.8 | 81.2 | 84.6 | 72.8 | 85.7 | 84.2 | 82.1 | 82.06 |
| | NLLB 1.3B | 66.4 | 72.0 | 69.2 | 69.20 | 81.3 | 81.9 | 84.2 | 68.5 | 84.5 | 84.2 | 83.1 | 81.10 |
| | NLLB 3.3B | 65.7 | 70.9 | 70.5 | 69.03 | 83.2 | 83.7 | 85.1 | 76.8 | 88.1 | 87.6 | 83.7 | 84.03 |

Table 4: Results of TTRAIN-ALL method on the target languages supported by the NLLB model(s) on AmericasNLI (accuracy) and NusaX (F1 score) with adapters and SFTs. The models are labelled by the number of parameters.

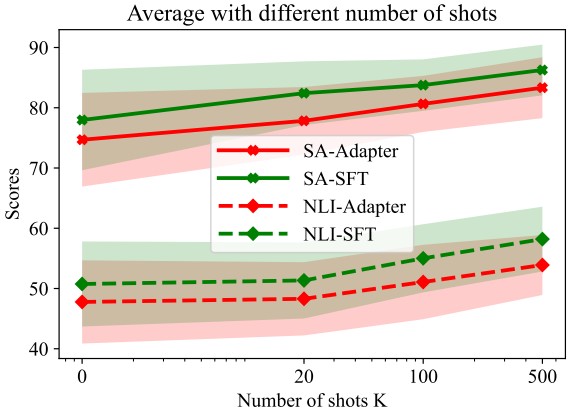

Figure 1: Performance when the number of gold-standard target shots K varies, taking values 0, 20, 100, and 500. We show the average across all target languages; the shaded area is the standard deviation (full results available in Appendix C).

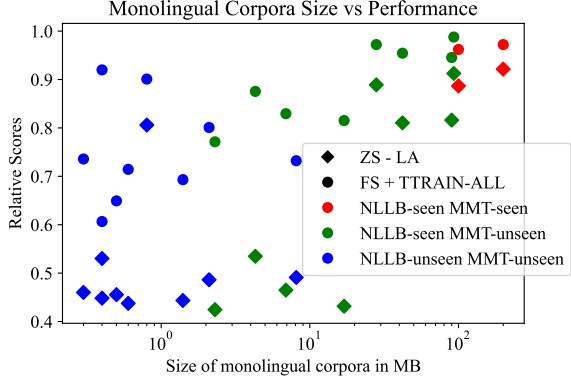

Figure 3: The scores of ZS - LA and FS + TTRAIN-ALL methods with SFTs against the monolingual corpus sizes. All target languages (from NLI and SA) are shown and they are grouped based on the coverage by the NLLB and MMT models. Relative scores are displayed (a fraction of the source language performance).

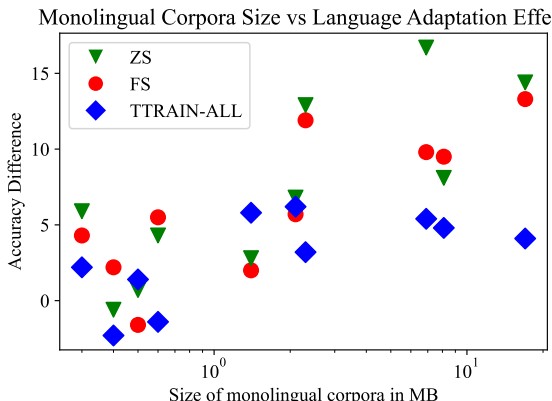

Figure 2: Gains from language adaptation against the size of monolingual corpora for NLI target languages. We show the difference in the accuracy of ZS (i.e. ZS - ZS − LA), FS and TTRAIN-ALL with adapters. Larger monolingual corpus size leads to larger gains when the language adaptation is present; the effect is more pronounced with ZS and FS than with TTRAIN-ALL.

TTRAIN-ALL configuration, language adaptation incurs gains of 3-4 points, while for ZS and FS they vary between 6-13 points. Figure 2 further illustrates this effect on all three methods with adapters, showing the gains from language adaptation on NLI languages against the size of their monolingual corpora: languages with larger corpora generally exhibit larger gains. The effect is visibly less pronounced for the TTRAIN-ALL where a relatively large amount of the target data (albeit translated) lessens the significance of language modules and monolingual corpora size. Finally, Figure 3 illustrates that the performance of our strongest configuration (FS + TTRAIN-ALL) with SFTs is the highest for languages that are seen by either the NLLB or MMT model – these are (incidentally) the languages with the largest corpora size too. Conversely, this pattern is absent with the ZS − LA configuration.

Language families for our target languages are given in Appendix A. While they could have an impact on the performance, it is difficult to disentangle the effect of the language family from the

amount of data available without having a much larger set of evaluation languages.

**PEFT Method.** The relative performances of the various configurations are very similar regardless of PEFT method, and in accordance with previous work (Ansell et al., 2022; Alabi et al., 2022), SFTs consistently outperform adapters by 2-3 points. However, we estimate that training adapters is generally around 3 times faster than training SFTs.

# 5 Related Work

**Parameter-Efficient Fine-Tuning Methods.** Parameter-efficient fine-tuning methods have emerged from a necessity to reduce compute and memory requirements of fine-tuning when dealing with large models. These methods can generally be grouped into those that modify the subset of parameters of the pretrained LLM (Ben Zaken et al., 2022; Guo et al., 2021; Sung et al., 2021) and those that introduce a completely fresh set of parameters to be updated (Li and Liang, 2021; Lester et al., 2021; Houlsby et al., 2019; Hu et al., 2022) allowing for different interactions with the pretrained model. They have been adopted for cross-lingual transfer as they are preferable when dozens of different fine-tunings for different languages and tasks need to be learned, stored and combined (Pfeiffer et al., 2020; Ansell et al., 2022; Parović et al., 2022). For a comprehensive overview of parameter-efficient fine-tuning, we refer the reader to Pfeiffer et al. (2023).

**Few-Shot Cross-Lingual Transfer.** Lauscher et al. (2020) and later Zhao et al. (2021) demonstrate the effectiveness of few-shot over the zero-shot cross-lingual transfer, showing that continued training of a *source-trained* model on a small number of labelled examples in the target language significantly increases performance (*target-adapting*). Schmidt et al. (2022) trade-off efficiency for performance by replacing the sequential fine-tuning procedure of Lauscher et al. (2020) with joint training on source and target language, showing it also improves training stability and robustness. They additionally show that first fine-tuning on multiple target languages provides extra performance gains. We adopt their joint training procedure, combining it further with parameter-efficient fine-tuning methods and language adaptation. Xu and Murray (2022) also exploit a joint source-and-target training procedure, further extending it to all target languages simultaneously instead of having language-

specific models, which becomes particularly attractive when dealing with a large number of target languages. They also introduce stochastic gradient surgery to circumvent the issue of conflicting gradients among languages.

Jundi and Lapesa (2022) compare few-shot and translation-based approaches, trying to gain an insight into which approach is better and under which circumstances. We consider these approaches in combination rather than in competition and find that the use of few-shot data can enhance performance even when a machine translation of the full source language dataset is available. However, their work complements ours by proposing a way to identify the examples which may be most profitable for humans to translate into target language "shots."

Winata et al. (2022) study few-shot cross-lingual transfer on languages unseen by MMTs using the NusaX dataset. They analyse the effectiveness of several few-shot strategies focusing on selecting languages for transfer and different learning dynamics exhibited by different types of MMTs.

**Machine Translation for Cross-Lingual Transfer.** The translate-train and translate-test approaches are common baselines for cross-lingual transfer (Conneau et al., 2020a; Hu et al., 2020). A number of enhancements have been proposed. Artetxe et al. (2020) showed that translate-test performance could be improved by training on back-translated rather than the original source language data to better model translation artefacts encountered at inference time. Ponti et al. (2021) note that translation-based approaches suffer from an error accumulation over the phases of the pipeline. They re-interpret this pipeline as a single model with an intermediate "latent translation" between the target text and its classification label, permitting the translation model to be fine-tuned according to a feedback signal from the task loss. Oh et al. (2022) show that the translate-train and translate-test approaches can be combined synergistically. Artetxe et al. (2023) show that translate-test is more favourable relative to translate-train than previously thought when better translation and monolingual models are used, and when measures are taken to correct the MT-induced mismatch between the data encountered at train and inference time. While we only consider applying translation-based cross-lingual transfer to classification tasks, prior work has considered its application to sequence labelling tasks as well (Jain et al., 2019; Fei et al., 2020;

García-Ferrero et al., 2022; García-Ferrero et al., 2022). For simplicity, we employ only continued pretraining on the standard MT task when adapting NLLB to unseen languages. Ko et al. (2021) enhance NMT model adaptation with additional tasks: denoising autoencoding, which exploits monolingual target language data; back-translation; and adversarial training which encourages the encoder to output language-agnostic features.

# 6 Conclusions and Future Work

We have investigated how to combine several cross-lingual transfer techniques which are applicable across several dimensions of resource scarcity into a single framework. We find that parameter-efficient language adaptation, few-shot learning and translate-train are complementary when employed in a multi-source training setup with few-shot upsampling. However, our training setup supports the use of any subset of these techniques depending on the availability of the necessary data and models. We remark on the significance of the finding that gold-standard few-shot target data can improve performance even when the entirety of the training data is translated into the target language by a high-quality NMT model. We also observe that languages not natively supported by an NMT model can benefit from translate-train through a simple adaptation procedure even with a small amount of parallel data.

## Limitations

Our experiments are based on two parameter-efficient fine-tuning methods: adapters and SFTs. This choice facilitates comparisons with the prior work in the area of cross-lingual transfer since these two methods have been studied extensively. However, we note that other modular and parameter-efficient fine-tuning methods are available and could be used in combination with our framework (Pfeiffer et al., 2023).

Our evaluation relies solely on classification tasks, as the data labels in these tasks are preserved upon translation into another language. This is not the case with sequence-labelling tasks, where an additional challenge lies in projecting the labels after obtaining the translation. Restricting our experiments to the classification tasks enables us to have a more controlled environment for studying only the effects of different data sources which is the main focus of this work. Studying other task

families could be done as part of future work.

During the adaptation of the MT model to unsupported languages, we only consider continued training with parallel data. While further performance increases could be achieved with the usage of monolingual data sources and backtranslation following Ko et al. (2021), we opt for simplicity, exploiting the monolingual data only with the language modules. Furthermore, this aligns with our goal which is not to maximize the raw performance but rather to study the effects of different data sources and their mutual interactions.

Due to a large number of experiments across many methods and ablations, we report all our results based on a single run. However, the large number of target languages we average over and the replication of the core findings across the two PEFT methods adds confidence that they are correct.

Finally, training language modules is typically computationally expensive. However, the modular design of cross-lingual transfer methods that we consider, enables us to train language modules only once and reuse them across all of our experiments.

## Acknowledgments

We would like to thank Mikel Artetxe for discussing this work with us and providing helpful advice. We also thank the anonymous reviewers for their comments.

Alan wishes to thank David and Claudia Harding for their generous support via the Harding Distinguished Postgraduate Scholarship Programme and Churchill College, Cambridge for travel assistance. Marinela Parović is supported by Trinity College External Research Studentship. Ivan Vulić acknowledges the support of a personal Royal Society University Research Fellowship *'Inclusive and Sustainable Language Technology for a Truly Multilingual World'* (no 221137; 2022–).

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

## A  Languages

The complete overview of languages, their codes and families, together with the monolingual data sizes and resources is provided in Table 5. The sizes and resources for the parallel corpora used in the MT model adaptation are given in Table 6.

## B  Ablation Experiments

We present per language results of our ablation experiments in Table 7. The summarised results are given in Table 3.

## C  Full Results with Different Number of Shots K

We give full results with the different number of gold-standard target shots K, where $K \in \{0, 20, 100, 500\}$. The setting $K = 0$ resembles the ZS approach, while the rest of the values fall within FS. The results are shown in Table 8, with their summary given in Figure 1.

| Task | Language | Code | Family | Corpus size (MB) | Corpus source(s) |
|------|----------|------|--------|------------------|------------------|
| Source | English | en | Indo-European, Germanic | 300,800 | Wikipedia |
| | Indonesian | id | Austronesian, Malayo-Sumbawan | 148,300 | |
| NLI | Aymara | aym | Aymaran | 2.3 | Tiedemann (2012); Wikipedia |
| | Asháninka | cni | Arawakan | 1.4 | Ortega et al. (2020); Cushimariano Romano and Sebastián Q. (2008); Mihas (2011); Bustamante et al. (2020) |
| | Bribri | bzd | Chibchan, Talamanca | 0.3 | Feldman and Coto-Solano (2020) |
| | Guarani | gn | Tupian, Tupi-Guarani | 6.9 | Chiruzzo et al. (2020); Wikipedia |
| | Náhuatl | nah | Uto-Aztecan, Aztecan | 8.1 | Gutierrez-Vasques et al. (2016); Wikipedia |
| | Otomí | oto | Oto-Manguean, Otomian | 0.4 | Hñähñu Online Corpus |
| | Quechua | quy | Quechuan | 17 | Agić and Vulić (2019); Wikipedia |
| | Rarámuri | tar | Uto-Aztecan, Tarahumaran | 0.6 | Brambila (1976) |
| | Shipibo-Konibo | shp | Panoan | 2.1 | Galarreta et al. (2017); Bustamante et al. (2020) |
| | Wixarika | hch | Uto-Aztecan, Corachol | 0.5 | Mager et al. (2018) |
| SA | Acehnese | ace | Austronesian, Malayo-Sumbawan | 90 | KoPI-NLLB (Cahyawijaya et al., 2022); LibriVox-Indonesia (Wirawan, 2022); NLLB-Seed (NLLB Team et al., 2022); Wikipedia |
| | Balinese | ban | Austronesian, Malayo-Sumbawan | 42 | INDspeech_NEWS_EthnicSR (Sakti and Nakamura, 2013), KoPI-NLLB (Cahyawijaya et al., 2022); LibriVox-Indonesia (Wirawan, 2022); NLLB-Seed (NLLB Team et al., 2022); Wikipedia |
| | Banjarese | bjn | Austronesian, Malayo-Sumbawan | 28 | KoPI-NLLB (Cahyawijaya et al., 2022); Korpus Nusantara (Sujaini, 2020); NLLB-Seed (NLLB Team et al., 2022); Wikipedia |
| | Buginese | bug | Austronesian, South Sulawesi | 4.3 | Korpus Nusantara (Sujaini, 2020); LibriVox-Indonesia (Wirawan, 2022); NLLB-Seed (NLLB Team et al., 2022); Wikipedia |
| | Javanese | jav | Austronesian, Javanese | 200 | Wikipedia |
| | Madurese | mad | Austronesian, Malayo-Sumbawan | 0.8 | Korpus Nusantara (Sujaini, 2020); Wikipedia |
| | Minangkabau | min | Austronesian, Malayo-Sumbawan | 93 | Indo Wiki Parallel Corpora (Trisedya and Inastra, 2014); KoPI-NLLB (Cahyawijaya et al., 2022); Korpus Nusantara (Sujaini, 2020); LibriVox-Indonesia (Wirawan, 2022); MinangNLP MT (Koto and Koto, 2020); Wikipedia |
| | Ngaju | nij | Austronesian, Barito | - | - |
| | Sundanese | sun | Austronesian, Malayo-Sumbawan | 100 | Wikipedia |
| | Toba Batak | bbc | Austronesian, Northwest Sumatra-Barrier Islands | 0.4 | Korpus Nusantara (Sujaini, 2020) |

Table 5: Details of the languages and monolingual data used for training and evaluation of SFTs and adapters. The corpora of Bustamante et al. (2020) are available at `https://github.com/iapucp/multilingual-data-peru`; all other NLI corpora mentioned are available at `https://github.com/AmericasNLP/americasnlp2021`; all the SA corpora (Cahyawijaya et al., 2022) are available through `https://indonlp.github.io/nusa-catalogue/`.

| Target Language | Source Language | Corpus size (#sent) | Corpus source(s) |
|-----------------|-----------------|---------------------|------------------|
| Bribri | Spanish | 8,502 | Feldman and Coto-Solano (2020) |
| Náhuatl | Spanish | 16,733 | Gutierrez-Vasques et al. (2016) |
| Otomí | Spanish | 5,488 | Hñähñu Online Corpus |
| Rarámuri | Spanish | 15,714 | Brambila (1976) |
| Shipibo-Konibo | Spanish | 15,586 | Galarreta et al. (2017); Bustamante et al. (2020) |
| Wixarika | Spanish | 9,960 | Mager et al. (2018) |
| Madurese | Indonesian | 629 | |
| Ngaju | Indonesian | 629 | Winata et al. (2023) |
| Toba Batak | Indonesian | 629 | |

Table 6: Details of the parallel corpora used for NLLB MT model adaptation. #sent = number of sentences in train + dev set.

| | Method | AYM | BZD | CNI | GN | HCH | NAH | OTO | QUY | SHP | TAR | **avg** |
|---|---|---|---|---|---|---|---|---|---|---|---|---|
| ADAPTER | ZS | 53.0 | 42.8 | 44.8 | 59.6 | 40.3 | 50.8 | 41.2 | 55.2 | 50.0 | 40.0 | 47.77 |
| | ZS – LA | 40.1 | 36.9 | 42.0 | 42.9 | 39.6 | 42.7 | 41.8 | 40.8 | 43.2 | 35.7 | 40.57 |
| | FS | 55.2 | 47.1 | 47.7 | 58.3 | 40.7 | 55.8 | 45.9 | 59.7 | 52.9 | 47.5 | 51.08 |
| | FS – LA | 43.3 | 42.8 | 45.7 | 48.5 | 42.3 | 46.3 | 43.7 | 46.4 | 47.2 | 42.0 | 44.82 |
| | FS – UPSAMPLE | 50.3 | 45.5 | 46.8 | 58.3 | 40.3 | 48.9 | **46.4** | 56.3 | 46.0 | 43.9 | 48.27 |
| | TTRAIN-ALL | **63.3** | **60.5** | **56.1** | **66.1** | **52.1** | **60.4** | 42.4 | **64.9** | **63.3** | 55.3 | **58.44** |
| | TTRAIN-ALL – LA | 60.1 | 58.3 | 50.3 | 60.7 | 50.7 | 55.6 | 44.7 | 60.8 | 57.1 | **56.7** | 55.50 |
| SFT | ZS | 58.4 | 44.7 | 47.6 | 62.2 | 44.4 | 50.8 | 46.4 | 60.4 | 49.5 | 43.1 | 50.75 |
| | ZS – LA | 35.7 | 38.7 | 37.3 | 39.1 | 38.3 | 41.3 | 37.7 | 36.3 | 40.9 | 36.8 | 38.21 |
| | FS | 59.2 | 58.7 | 53.2 | 63.9 | 45.9 | 54.6 | 49.1 | 61.2 | 53.1 | 51.2 | 55.01 |
| | FS – LA | 53.3 | 54.9 | 47.6 | 48.9 | 42.9 | 53.8 | 45.5 | 53.3 | 46.5 | 48.4 | 49.51 |
| | FS – UPSAMPLE | 59.1 | 51.2 | 50.3 | 63.9 | 42.9 | 52.7 | 48.1 | 62.8 | 50.1 | 44.9 | 52.60 |
| | TTRAIN-ALL | 65.5 | 61.9 | 56.7 | 70.5 | 55.5 | 62.2 | 42.6 | 68.5 | 66.0 | 58.7 | 60.81 |
| | TTRAIN-ALL – LA | 63.6 | 60.7 | 48.0 | 65.6 | 54.7 | 57.3 | 46.9 | 63.6 | 62.7 | 55.7 | 57.88 |

(a) AmericasNLI: accuracy

| | Method | ACE | BAN | BBC | BJN | BUG | JAV | MAD | MIN | NIJ | SUN | **avg** |
|---|---|---|---|---|---|---|---|---|---|---|---|---|
| ADAPTER | ZS | 74.9 | 78.0 | 72.3 | 77.6 | 57.6 | 82.9 | 68.5 | 79.9 | – | 80.5 | 74.69 |
| | ZS – LA | 68.0 | 70.8 | 37.6 | 78.3 | 31.9 | 80.9 | 63.5 | 77.9 | 66.9 | 78.0 | 65.38 |
| | FS | 79.1 | 80.5 | 77.2 | **86.0** | 72.0 | 85.0 | 77.5 | **84.8** | – | 83.7 | 80.64 |
| | FS – LA | 74.7 | 73.5 | 64.3 | 79.6 | 61.7 | 82.3 | 74.0 | 81.5 | 70.9 | 77.8 | 74.03 |
| | FS – UPSAMPLE | 76.5 | 78.4 | 71.9 | 80.6 | 64.5 | 82.5 | 74.6 | 80.1 | – | 81.9 | 76.78 |
| | TTRAIN-ALL | **79.6** | **82.3** | **81.0** | 85.0 | 68.1 | 86.2 | 78.8 | 83.3 | – | **85.7** | **81.11** |
| | TTRAIN-ALL – LA | 78.5 | 76.5 | 73.1 | 83.2 | 67.6 | 82.7 | 77.8 | 82.0 | **77.4** | 79.5 | 77.83 |
| SFT | ZS | 80.0 | 81.3 | 65.8 | 82.0 | 63.8 | 84.3 | 73.5 | 86.6 | – | 84.4 | 77.97 |
| | ZS – LA | 72.8 | 72.3 | 47.3 | 79.3 | 47.7 | 82.2 | 71.9 | 81.4 | 70.3 | 79.1 | 70.43 |
| | FS | 82.2 | 84.1 | 80.6 | 88.3 | 77.7 | 88.0 | 78.6 | 89.2 | – | 85.1 | 83.76 |
| | FS – LA | 78.9 | 76.2 | 68.6 | 81.1 | 71.0 | 86.3 | 78.2 | 84.0 | 76.0 | 82.9 | 78.32 |
| | FS – UPSAMPLE | 82.7 | 84.5 | 79.5 | 87.9 | 74.7 | 86.5 | 78.0 | 87.8 | – | 87.0 | 83.18 |
| | TTRAIN-ALL | 83.4 | 82.4 | 82.7 | 84.6 | 76.8 | 86.4 | 81.6 | 87.4 | – | 85.2 | 83.39 |
| | TTRAIN-ALL – LA | 78.9 | 81.0 | 75.1 | 83.0 | 67.1 | 86.2 | 79.7 | 82.8 | 77.6 | 82.5 | 79.39 |

(b) NusaX: F1

Table 7: Per-language results of ablation experiments with ZS, FS, and TTRAIN-ALL methods on NLI and SA with adapters and SFTs: – LA denotes the absence of language adaptation (i.e. only the task module is present), while – UPSAMPLE indicates there is no upsampling of the gold-standard target shots.

| | Method | AYM | BZD | CNI | GN | HCH | NAH | OTO | QUY | SHP | TAR | **avg** |
|---|---|---|---|---|---|---|---|---|---|---|---|---|
| ADAPTER | ZS (K = 0) | 53.0 | 42.8 | 44.8 | 59.6 | 40.3 | 50.8 | 41.2 | 55.2 | 50.0 | 40.0 | 47.77 |
| | FS (K = 20) | 50.4 | 43.3 | 46.0 | 56.9 | 39.9 | 50.4 | 43.2 | 58.3 | 50.5 | 44.0 | 48.29 |
| | FS (K = 100) | 55.2 | 47.1 | 47.7 | 58.3 | 40.7 | 55.8 | 45.9 | 59.7 | 52.9 | 47.5 | 51.08 |
| | FS (K = 500) | **57.2** | **54.0** | **53.7** | **61.6** | **45.1** | 55.1 | **48.0** | 59.1 | **54.7** | 50.5 | **53.90** |
| SFT | ZS (K = 0) | 58.4 | 44.7 | 47.6 | 62.2 | 44.4 | 50.8 | 46.4 | 60.4 | 49.5 | 43.1 | 50.75 |
| | FS (K = 20) | 57.9 | 48.7 | 46.4 | 59.9 | 44.3 | 51.5 | 48.7 | 61.2 | 51.1 | 43.7 | 51.34 |
| | FS (K = 100) | 59.2 | 58.7 | 53.2 | 63.9 | 45.9 | 54.6 | 49.1 | 61.2 | 53.1 | 51.2 | 55.01 |
| | FS (K = 500) | 61.1 | 59.5 | 55.7 | 65.6 | 48.7 | 58.9 | 52.5 | 64.4 | 61.5 | 54.1 | 58.20 |

(a) AmericasNLI: accuracy

| | Method | ACE | BAN | BBC | BJN | BUG | JAV | MAD | MIN | SUN | **avg** |
|---|---|---|---|---|---|---|---|---|---|---|---|
| ADAPTER | ZS (K = 0) | 74.9 | 78.0 | 72.3 | 77.6 | 57.6 | 82.9 | 68.5 | 79.9 | 80.5 | 74.69 |
| | FS (K = 20) | 74.6 | 78.9 | 73.6 | 81.0 | 66.1 | 84.3 | 78.3 | 81.3 | 82.4 | 77.83 |
| | FS (K = 100) | 79.1 | 80.5 | 77.2 | 86.0 | 72.0 | 85.0 | 77.5 | 84.8 | 83.7 | 80.64 |
| | FS (K = 500) | **81.1** | **84.2** | **78.5** | **88.0** | **74.8** | **87.9** | **79.4** | **88.0** | 88.1 | **83.33** |
| SFT | ZS (K = 0) | 80.0 | 81.3 | 65.8 | 82.0 | 63.8 | 84.3 | 73.5 | 86.6 | 84.4 | 77.97 |
| | FS (K = 20) | 81.9 | 83.1 | 79.0 | 87.9 | 73.4 | 86.3 | 76.0 | 87.0 | 87.2 | 82.42 |
| | FS (K = 100) | 82.2 | 84.1 | 80.6 | 88.3 | 77.7 | 88.0 | 78.6 | 89.2 | 85.1 | 83.76 |
| | FS (K = 500) | 86.5 | 89.3 | 83.1 | 87.9 | 79.7 | 91.3 | 80.6 | 90.5 | 87.5 | 86.27 |

(b) NusaX: F1

Table 8: Per-language results on NLI and SA with the different number of gold-standard target shots K. We consider ZS (K = 0) and FS with $K \in \{20, 100, 500\}$ with adapters and SFTs.