# OpenReview forum: "Unifying Cross-Lingual Transfer across Scenarios of Resource Scarcity"
_EMNLP/2023/Conference — EMNLP 2023 Main_

### Official Review · Reviewer_8j5G · 2023-08-05

**Soundness:** 4

**Excitement:**

4: Strong: This paper deepens the understanding of some phenomenon or lowers the barriers to an existing research direction.

**Paper Topic And Main Contributions:**

This paper combines a number of techniques designed to improve cross-lingual transfer for low-resource languages into a single experimental framework.

1) Few shot learning as an improvement to zero-short. Noting how effective a small amount of gold data is, the authors further elaborate on the methodology by upsampling the few-shot data compared to other generated training data.

2) Parameter-efficient finetuning  (e.g., adaptor networks) to "adapt" base multilingual LLMs to a new language.

3) Translate-train methods to produce synthetic training data in the low-resource language via NMT translation from a higher-resource language. NMT systems can themselves be adapted by additional training on even a small amount of parallel data in the low-resource language.

All of the methods add some performance to a downstream NLI task, but the experiments in this paper show that the benefits are complementary/additive - each additional method used adds more performance, so as many as possible should be used as resources allow.



**Reasons To Accept:**

Presents empirical evidence that nearly all the low-resource strategies for cross-lingual transfer learning scattered across the literature are viable (at least in the downstream NLI/classification task) and complementary so any available combination will provide additive performance benefits. Similarly, a small amount of gold "real" data remains valuable even when a large amount of synthetic data is available.

**Reasons To Reject:**

The individual methods used in the paper are mostly off-the-shelf (although the paper makes some minor methodological adjustments noted above). Additive performance benefits are not very surprising, since there is no clear way in which one method would undo another.

**Reproducibility:**

4: Could mostly reproduce the results, but there may be some variation because of sample variance or minor variations in their interpretation of the protocol or method.

**Reviewer Confidence:**

4: Quite sure. I tried to check the important points carefully. It's unlikely, though conceivable, that I missed something that should affect my ratings.

---

> ### Author Rebuttal · Authors · 2023-08-29
>
> Thank you for your encouraging review!
>
> While we agree that the contributions are primarily empirical rather than methodological, we would argue that the way we have unified the various “off-the-shelf” methods is not trivial and does not appear to have been attempted before in the literature. It is not obvious how complementary different tools and data sources will be, nor how much improvements they could give when combined. For instance, the finding that large amounts of machine-translated data cannot eliminate the utility of human-crafted examples is a useful, and in our opinion non-obvious, finding. In general, we hope that the empirical contributions of our work will provide a set of useful reference points for future research on the integration of different cross-lingual transfer techniques.

---

### Official Review · Reviewer_jn1b · 2023-08-06

**Soundness:** 4

**Excitement:**

3: Ambivalent: It has merits (e.g., it reports state-of-the-art results, the idea is nice), but there are key weaknesses (e.g., it describes incremental work), and it can significantly benefit from another round of revision. However, I won't object to accepting it if my co-reviewers champion it.

**Paper Topic And Main Contributions:**

This paper conducts a comprehensive study combining different cross-lingual techniques and applying them across several dimensions of resource scarcity. This study provides insights into the complementarity of different data used to bridge the gap between resource-rich and resource-scarce languages. More specifically, their analysis investigates the effectiveness of the resourcefulness of various data sources employed at different stages of the transfer: pre-training and fine-tuning either zero-shot or few-shot manner or using data augmentation in the form of machine-translated data. All this provides a valuable toolbox to integrate different transfer learning strategies and adapt them to different scenarios.

**Questions For The Authors:**

- Why have you chosen to evaluate on those datasets and languages?
- Where are studies of different languages with respect to different degrees of coverage in the pre-trained model?

**Reasons To Accept:**

- It is nice to see a study that provides a comprehensive analysis spanning different dimensions of resourcefulness in transfer learning which a lot of studies have focused on separately.
- There are a lot of generalizable insights and recommendations which can be used in using multiple transfer learning techniques
- It is useful to understand the complementarity of different techniques of resourcefulness


**Reasons To Reject:**

- There are a lot of techniques not explored and the study is more focused on parameter-efficient language models only. Analyzing also the biases in the resourcefulness of those language models which are believed to have an equally if not higher impact on the outcome of the transfer is missing.
- It is a bit hard to get a hand on different experiments performed and get a full picture. It would help to outline those experiments in a clearer manner in the experimental setup (see presentation improvements below).
- The study sounds a bit incremental. Instead of adopting different techniques why not just have a more holistic unbiased analysis.

**Reproducibility:**

4: Could mostly reproduce the results, but there may be some variation because of sample variance or minor variations in their interpretation of the protocol or method.

**Reviewer Confidence:**

3: Pretty sure, but there's a chance I missed something. Although I have a good feel for this area in general, I did not carefully check the paper's details, e.g., the math, experimental design, or novelty.

**Typos Grammar Style And Presentation Improvements:**

Abstract:

- Line 011-013: it would be helpful to list those prior work and compare them with your work by highlighting which specific dimensions have they lacked in their evaluation and maybe summarizing that in some table or figure that you can show in the first page. I see some overalp with large-scale performance evaluation works like XTREME and XTREME-R but you don't discuss those differences in details in the introduction.

- Section 3: I think this could benefit from some diagrams to explain the strategies to evaluate the complementarity of different techniques planned.
- Line 382: Table 3 should be either talked about right here or shown later on in language resourcefulness paragraph
- Table 2: doesn't make sense to compare zero-shot, few-shot, translation etc. they all use different levels of resources. it would be more interesting to have correlation analysis where you assess the improvements across different techniques but while fixing the amount of resourcefulness and define some measure to do that.
- Hard to read Table 2 with two datasets with so many numbers. I think it would be better if bar plots are used or the differences in the performances are highlighted for more visual easiness.
- Figure 2: small x and y axis labels

---

> ### Author Rebuttal · Authors · 2023-08-29
>
> Thank you for your review and encouraging feedback!
>
> Clarifications and responses to the reasons to reject:
> * We focus on parameter-efficient cross-lingual transfer methods as they have been shown to be the most effective ways of transfer to low-resource and unseen languages which is where all our languages land (18/20 are unseen by the MMT) - see the original MAD-X and SFT papers. There is not much bias due to the resourcefulness of target languages during MMT pre-training because most of the languages we consider are unseen by the MMT. However, we perform some analysis of this type in Figure 3, where we can see from the red group of symbols that the languages seen during pre-training are among the best-performing initially, but they still benefit from LA/FS/TTRAIN.
> * We give the outline of our experiments in section 3.3 Configurations and Ablations. However, we will use the extra space to make this clearer, and we appreciate your presentation suggestions which could also help with this.
> * We are a bit perplexed by your suggestion that the study should be more holistic, given that it unifies a number of approaches that have not been considered together before across a range of cross-lingual transfer scenarios, and one of the main goals of this study has been indeed to provide a more holistic approach to cross-lingual transfer that integrates different components researched and applied previously in isolation. We are also unclear on what you mean by "unbiased" in this context.
>
> Clarifications and responses to the questions:
> * We justify our choice of the datasets in L252-256. We focus on low-resource, classification benchmarks that have sufficiently large train/validation splits for few-shot learning to simulate different resource levels. We wanted the datasets to have languages with a range of resource levels within the low-resource umbrella, i.e. seen or unseen by the MMT/translation model and various corpus sizes.
> * This analysis is provided in the paragraph Language Resourcefulness (L450-472) and Figure 3.
>
> In response to your presentation improvements:
> * We provide a detailed overview of the related work in Section 5, with the subheadings denoting the class of cross-lingual transfer approach considered - generally, there is no mixing of different approaches within these papers. XTREME and XTREME-R are examples of treating these approaches separately, without utilizing any monolingual corpora for further improvements.
> * We restrict the analysis of varying corpora size across different languages rather than varying it within a single language - while this would be a valid and interesting experiment, we prioritized coverage over more languages to improve the generality of our findings. On the other hand, we do provide analysis on varying the number of gold-standard target shots.
> * We will take into account your suggestions on mentioning Table 3 later in the text, presenting the results with the bar charts, and increasing the font on the plots. Thank you very much for the suggestions - it should be possible to further smooth the presentation details given one extra page of content!

---

### Official Review · Reviewer_TbEt · 2023-08-11

**Soundness:** 4

**Excitement:**

4: Strong: This paper deepens the understanding of some phenomenon or lowers the barriers to an existing research direction.

**Paper Topic And Main Contributions:**

The paucity of data across numerous global languages mandates the transference of knowledge from those languages endowed with abundant resources. Historically, the dimensions and techniques employed to address this issue have been handled in isolation. In contrast, this paper presents a more synthesized perspective, investigating the application of an extensive suite of cross-lingual transfer tools across various scenarios, with particular emphasis on the most exigent, low-resource situations.

**Questions For The Authors:**

It would be beneficial for the authors to provide analyses on:
- The effect of varying corpus sizes;
- The impact related to different language families;
- The influence of distinct language types (e.g., SVO, SOV...) within the context of the transfer-learning approach.

**Reasons To Accept:**

This paper is an essential contribution to the field as it innovatively illustrates the unification of various cross-lingual transfer techniques across different dimensions of resource scarcity, all within a single cohesive framework. What sets this work apart is the discovery that parameter-efficient language adaptation, few-shot learning, and translate-train approaches are not just parallel strategies but are in fact complementary. When these are harmoniously integrated within a multi-source training setup coupled with few-shot upsampling, the synergy leads to a more robust and efficient model. The paper's results bear significant implications for the future of language processing, and the techniques proposed have the potential to reshape the way we approach multilingual challenges in resource-constrained environments.

**Reasons To Reject:**

No.

**Reproducibility:**

4: Could mostly reproduce the results, but there may be some variation because of sample variance or minor variations in their interpretation of the protocol or method.

**Reviewer Confidence:**

4: Quite sure. I tried to check the important points carefully. It's unlikely, though conceivable, that I missed something that should affect my ratings.

---

> ### Author Rebuttal · Authors · 2023-08-29
>
> Thank you for your very encouraging review, we really appreciate it!
>
> In answer to your questions on additional analyses:
> * We provide a study on the effect of varying corpus sizes in the paragraph Language Resourcefulness (L450-472) and Figures 2 and 3, showing that: (i) languages with larger corpora exhibit larger gains from language adaptation; (ii) languages seen by MMT and/or MT models which are also languages with the largest corpora obtain the highest overall performance.
> * We will use the extra space to include a paragraph describing the impact of different language families.
> * Previous work has shown that the word order of source and target languages has an impact on the cross-lingual transfer (K et al. Cross-Lingual Ability of Multilingual BERT: An Emprical Study [ICLR 2020]; Pires et al. How multilingual is Multilingual BERT? [ACL 2019]), so there is a reason to believe different language types could affect the transfer gap in our experiments too. However, we see such analysis as beyond the scope of this work whose aim is to integrate various cross-lingual transfer tools and evaluate their performances in synergy.
>
> Regarding reproducibility, we will release the code which will make our findings easily reproducible.

---

### Meta-Review · Area_Chair_D1v4 · 2023-09-17

**Recommendation:** 5

**Metareview:**

This paper conducts a comprehensive study on how several approaches work together for very low-resource languages including few-shot learning, parameter-efficient tuning, and translate-train methods. The paper finds that these methods could be complementary and offer recommendations on how to combine them together. The experiments are comprehensive as well.

---

### Decision · Program_Chairs · 2023-10-07

**Decision:**

Accept-Main

**Comment:**

This paper conducts a comprehensive study on how several approaches work together for very low-resource languages including few-shot learning, parameter-efficient tuning, and translate-train methods. The paper finds that these methods could be complementary and offer recommendations on how to combine them together. The experiments are comprehensive as well.